# Metacognitive Retrieval-Augmented Large Language Models

## ABSTRACT

Retrieval-augmented language models have become central in natural language processing due to their efficacy in generating precise and relevant content. While traditional methods employ single-time retrieval, more recent approaches have shifted towards multi-time retrieval for complex, multi-hop reasoning tasks. However, current strategies, despite their advancements, are bound by predefined reasoning steps, potentially leading to inaccuracies in response generation. This paper introduces the Metacognitive Retrieval-Augmented Generation framework (MetaRAG), a novel approach that combines the retrieval-augmented generation process with human-inspired metacognition. Drawing from cognitive psychology, metacognition allows an entity to self-reflect and critically evaluate its cognitive processes. By integrating this, MetaRAG enables the model to monitor, evaluate, and plan its response strategies, enhancing its introspective reasoning abilities. Through a three-step metacognitive regulation pipeline, the model assesses the adequacy of its answers, identifies reasons for potential inadequacies, and formulates plans for refinement. Empirical evaluations on multi-hop QA datasets show that MetaRAG significantly outperforms existing methods.

## KEYWORDS

Retrieval-Augmented Generation, LLMs, Metacognition

## 1 INTRODUCTION

Recently, large language models (LLMs) have emerged as a foundational component in various natural language processing tasks, attributed to their remarkable capability to comprehend and generate human-like language [23]. While these models are endowed with vast repositories of knowledge learned during training, they exhibit the propensity to generate hallucinated content [21, 39]. To address this issue, researchers have introduced the idea of integrating retrieval systems into LLMs. By doing so, LLMs can look up relevance information from external knowledge bases, ensuring a more reliable and precise content generation.

Historically, retrieval-augmented language models [9–11, 20] have primarily employed single-time retrieval, extracting knowledge once based on an initial query. This method, while effective for tasks with straightforward informational needs, falls short when faced with complex tasks demanding multi-faceted information or multi-step reasoning. Recognizing this limitation, recent researches have shifted towards a multi-time retrieval framework [1, 15, 25, 33]. This method doesn't confine knowledge retrieval to one instance

*Conference'17, July 2017, Washington, DC, USA*
© 2023 Association for Computing Machinery.
ACM ISBN 978-x-xxxx-xxxx-x/YY/MM. . . $15.00
https://doi.org/10.1145/nnnnnnn.nnnnnnn

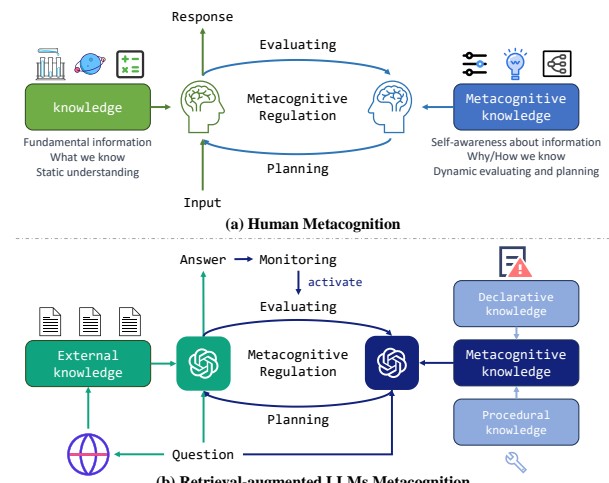

**Figure 1: The correspondence between the metacognitive processes in humans and retrieval-augmented LLMs**

but revisits it iteratively during the generation process, but decomposing the primary question into sub-questions [16], or leveraging partially generated content [24, 37] and forward-looking sentences [12] as dynamic search queries.

Although previous methods have made strides in improving the quality of generated answers, they strictly adhere to predefined reasoning steps over all questions. Such inflexible approaches lack the ability to diagnose specific errors in their responses and consequently don't possess mechanisms to enhance their performance. We argue that this limitation might stem from the model's lack of awareness regarding its own reasoning processes. When humans confront complex issues, they often reflect on their thought patterns, gradually adjusting and optimizing their strategies. This ability comes from our innate **metacognition**, which enables introspection, self-assessment, and self-regulation. Inspired by it, we aim to integrate metacognitive ability into LLMs to enhance retrieval-augmented generation (RAG). By adopting this approach, the model is able to identify its own inaccuracies and dynamically adjust their reasoning strategies, leading to more precise answer generation.

Derived from the field of cognitive psychology [19, 27], metacognition concerns an individual's capacity to self-reflect and critically evaluate their cognitive processes [34]. As shown in the Figure 1(a), it can be classified into two integral components: metacognitive knowledge and metacognitive regulation. The former refers to an individual's self-awareness of their cognitive strengths, limitations, and mechanisms. On the other hand, metacognitive regulation [4] involves the active management and control of one's cognitive processes. Empowered by metacognitive capabilities, the human brain possesses the capacity to discern the underlying rationale behind responses and acquire the means for self-improvement.

Drawing inspiration from human metacognitive processes, we introduce the Metacognitive Retrieval-Augmented Generation framework (MetaRAG). As illustrated in Figure 1(b), MetaRAG features a

"cognition-metacognition" collaborative framework. The cognition component is responsible for deriving answers from the provided question and references, while the metacognitive component, acting as a critic model, delves deep into potential mistakes during reasoning. Upon conducting an analysis of model performance under different conditions of knowledge (as detailed in Sec. 3.2), it has been observed that there are three main reasons causing the model fails to infer the correct answer: **insufficient knowledge**, **conflicting knowledge**, and **erroneous reasoning**. Endowed with the benefit of metacognitive mechanism, we expect the model to be aware of its own cognitive process in RAG tasks from two aspects: (1) The sufficiency and harmonization of external retrieved knowledge and LLM's intrinsic knowledge. (2) The reliability and accuracy of multi-hop reasoning. By doing so, the model is capable of identifying potential issues present in knowledge integration and answer reasoning, thereby enabling targeted improvements.

Specifically, we delineate the metacognitive process into three distinct steps in the context of retrieval-augmented LLMs. (1) **Monitoring** assesses the quality of the current response to determine whether there's a need to invoke the metacognitive evaluating. (2) **Evaluating** is to identify the reasons why the current answer may not meet the requirements. During this phase, the model leverages metacognitive knowledge to analyze the flaws in the response. This knowledge encompasses two main areas: declarative knowledge, which involves recognizing prevalent error patterns, and procedural knowledge, focusing on the utilization of methods to assess the sufficiency and harmonization of both internal and external knowledge. Based on this evaluation, results are categorized into four distinct scenarios. (3) **Planning** offers tailored suggestions for the cognitive component on potential improvements. For each of the aforementioned scenarios in the evaluating stage, distinct planning strategies are designed to enhance the original cognitive process. Collectively, these steps ensure that the model not only identifies inadequacies in its initial cognitive responses but also fixes them based on the metacognitive evaluating and planning. The experimental results on two multi-hop question answering (QA) datasets indicate that MetaRAG gains higher capabilities of reasoning and outperforms existing baselines significantly.

The contributions of this paper are summarized as: (1) We introduce a metacognitive retrieval-augmented generation framework that integrates LLMs with human introspective reasoning for multi-hop QA tasks. (2) Through empirical analysis, we summarize three primary challenges in multi-hop QA causing wrong answers: insufficient knowledge, conflicting knowledge, and erroneous reasoning. (3) We devise a three-step metacognitive regulation pipeline tailored for retrieval-augmented LLMs, offering a systematic way for models to assess, diagnose, and refine the original cognitive process.

## 2 RELATED WORK

The development of retrieval-augmented language models has been a central theme in recent research endeavors. Their aim is to harmoniously marry the static knowledge encapsulated within the language model to the dynamic wealth of information on the web. The development of these models can be bifurcated into two primary phases: single-time retrieval and multi-time retrieval.

**Single-time Retrieval**. Initial endeavors in retrieval-augmented language models predominantly embraced the single-time retrieval strategy [6, 9, 20, 38]. In this framework, a single-time extraction of knowledge was performed in response to the user's initial query. Various methods were conceived to incorporate this external knowledge retrieval. For instance, Guu et al. [5] introduced a language model that incorporated latent knowledge retrieval during pre-training, whereas Ram et al. [25] chose to keep the core LM architecture untouched, simply appending grounding documents to its input. Meanwhile, Shi et al. [29] perceived the language model as an inscrutable entity, complementing it with an externally trainable retrieval module. Such strategies showcased remarkable efficacy for tasks that demanded straightforward information, like factoid question answering [18] and fact verification [31]. However, their applicability waned for intricate tasks demanding multi-hop reasoning, as the single-time retrieval lacked the depth to decode the subtleties embedded in complex inquiries.

**Multi-time Retrieval**. To counter the shortcomings of the single-time retrieval paradigm, the spotlight shifted towards the development of multi-time retrieval models. This paradigm champions an iterative knowledge extraction process throughout content generation. Some approaches [15, 22, 33] are designed to passively harness past contexts, conducting retrievals at predetermined intervals. Others [17, 37] deconstruct a multifaceted query into a series of simpler sub-queries, each necessitating its distinct retrieval operation. Furthermore, the intrinsic capabilities of the latest LLMs have been harnessed to autonomously dictate the timing and content of retrievals. For instance, Press et al. [24] leverages the model's partially generated content as evolving search queries, allowing it to iteratively refine its search. Meanwhile, Jiang et al. [12] strategically uses prospective sentences as dynamic search triggers. The ReAct model [37] ingeniously fuses a Chain-of-Thought (CoT) rationale with action in a seamless thought-action-observation loop. Other innovative approaches [28, 30] embed introspective mechanisms that iteratively refine the model's outputs. This iterative retrieval approach proves effective for queries with inherent ambiguities or those demanding a synthesis of diverse information sources.

In contrast to the aforementioned studies, this paper conducts an exploration of The fundamental reasons for causing the model to answer incorrectly in RAG. Drawing inspiration from the domain of cognitive psychology, we integrate metacognitive ability into LLMs to enable the model to be aware of its reasoning process, thus enhancing the quality of answer generation.

## 3 PRELIMINARY

In this section, we formulate the task of retrieval-augmented generation and investigate its limitations on multi-hop QA.

### 3.1 Task Definition

Given a question $q$ and a retrieval corpus $D = \{d_i\}_{i=1}^{|D|}$ (with Wikipedia articles serving as the primary data source in this study), the goal of retrieval-augmented LLMs is to generate an answer $y$ based on the question as well as the documents retrieved in relation to it. This can be represented as:

$$y = \text{LLM}_{\text{QA}}([D_q, q], \text{Prompt}_{\text{QA}}), \tag{1}$$

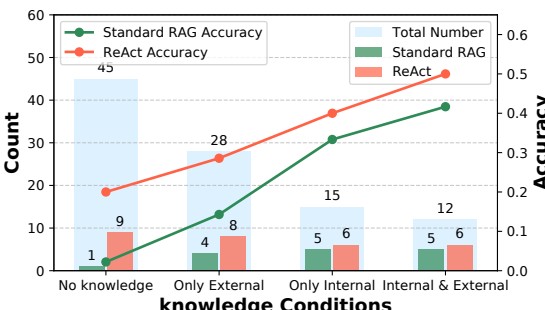

**Figure 2: Comparisons of single- and multi-time retrieval-augmented LLMs under different conditions of knowledge.**

Prompt$_{QA}$: Please act as a question-answering system, answer the {question} based on the {retrieved documents}

where $D_q$ is the retrieved documents for the query $q$, which is a set of Wikipedia articles retrieved by BM25 [26] in our experiments. $[\cdot, \cdot]$ is concatenation following designated prompts. LLM$_{QA}$ is the role of the LLM, which concentrates on question answering tasks.

### 3.2 Task Exploration

We conduct an empirical study to evaluate the effectiveness of retrieval-augmented LLMs under various knowledge conditions. Our main aim is to ascertain whether a question can be answered utilizing either the intrinsic knowledge within the LLM or via externally retrieved documents. Through human annotation of (a) the quality of closebook answers from the LLM and (b) the knowledge completeness of retrieved documents, we are able to categorize questions into four distinct conditions:

- *No knowledge.* Neither the LLM nor retrieved documents can provide an answer correctly.
- *Only external.* Answers can be found in retrieved documents, but not directly from the LLM.
- *Only internal.* The LLM can answer the question directly, but external documents cannot provide the solution.
- *Both internal and external.* The question can be addressed either directly by the LLM or through retrieved documents.

For comparison, both standard RAG [2] and ReAct [37] are put to the test on sampled 100 questions from HotpotQA dataset.

The diagram in Figure 2 depicts the efficacy of various methods. Our study yields several insights: (1) When the model operates without any knowledge, it faces difficulty in generating accurate responses. Nevertheless, the ReAct method, with multi-time retrieval mechanism, enhances the accuracy for such queries. (2) When the model is based only on either internal or external knowledge, there's a noticeable accuracy improvement. However, conflicts in the knowledge limit the model's ability to answer questions correctly. The ReAct method doesn't sufficiently mitigate this issue. (3) In situations where both internal and external knowledge sources concur and can tackle the question at hand, there's a marked improvement in the model's accuracy. Yet, it still isn't flawless. This suggests that even with complete knowledge, the model can err due to incorrect reasoning. These insights highlight three primary challenges in multi-hop QA when the model fails to answer correctly: **insufficient knowledge**, **conflicting knowledge**, and **erroneous**

**reasoning**. In subsequent sections, we will delve deeper into how metacognitive strategies can help overcome these challenges.

## 4 METACOGNITIVE RAG

Retrieval augmentation has become one of the primary methods to mitigate the hallucination issues in LLMs. However, existing research on retrieval-augmented LLMs primarily focuses on the design of the reasoning steps, overlooking the awareness of the reasoning itself. Motivated by this observation, in this section, we introduces a metacognitive retrieval-augmented generation framework. This approach taps into the principles of metacognition, allowing for introspection of the cognitive process. By doing so, it identifies shortcomings in the reasoning process and aims to enhance the accuracy of answer derivation.

The overall framework of MetaRAG is depicted in Figure 3. MetaRAG comprises two spaces: the cognition space and the metacognition space. The former functions as a QA system, while the latter serves as both an evaluator and critic, introspecting the reasoning process. This metacognition space primarily encompasses three main phases: (1) **Monitoring**; (2) **Evaluating**; (3) **Planning**. The following sections introduce the details of these three steps.

### 4.1 Monitoring: Assessing Answer Satisfaction

The primary function of monitoring is to keep track of one's cognitive processes. In human brain, not all cognitive activities necessarily trigger metacognitive evaluating [19]. Typically, only when the problem becomes so complex that the correctness of the cognitive process cannot be guaranteed, it becomes necessary to "think thrice before answering". In multi-hop QA tasks, due to the complexity of the task or insufficient knowledge, retrieval-augmented LLMs sometimes fail to reason out the correct answer. The role of monitoring is to assess the satisfaction of the answer, which then determines whether to activate the metacognitive evaluating phase.

To investigate the conditions under which an answer is deemed satisfactory, we hypothesize that an answer is highly plausible when the cognition of the LLM aligns with the cognition of an expert model. Conversely, certain deviation necessitates the intervention of metacognition. With this in mind, we select an expert model on QA tasks to evaluate the satisfaction level of the answers produced. Specifically, given a question $q$, retrieved documents $D_q$, we first prompt the expert model $M$ to generate an answer:

$$y' = M_\phi([D_q, q]), \qquad (2)$$

where $\phi$ is the parameters of the model $M$. Next, we decide on the model's subsequent action by computing the similarity between the LLM outputs $y$ and the expert model's outputs $y'$. The decision of the next action is defined as:

$$\text{Action} = \begin{cases} \text{Activate evaluating stage} & \text{if } \langle \vec{X}_y, \vec{X}_{y'} \rangle < \text{k}, \\ \text{Output the answer} & \text{otherwise.} \end{cases}$$

Here, $k$ serves as a threshold value governing the model's behavior. A higher value of $k$ implies that a greater number of reasoning processes require metacognitive evaluation. $\vec{X}$ represents embeddings encoded by an encoder (e.g. BERT Encoder), and $\langle, \rangle$ is the similarity function, implemented by cosine similarity. In cases where the similarity between the LLM output and the expert model output falls

**Figure 3: The overall architecture of MetaRAG. The green part is the cognition space, focusing on reasoning the answer for a given question. In contrast, the blue part is the metacognition space, responsible for monitoring, evaluating, and planning.**

below a certain threshold, the expert model triggers metacognitive process, including metacognitive evaluating and planning.

## 4.2 Evaluating: Identifying Answer Limitations

When a monitor discerns that an answer fails to fully address a question, it triggers the metacognitive process of evaluating. This introspective exercise is geared towards identifying the shortcomings of the provided response and discerning why the model may have faltered in its reasoning. Central to this introspection are two pivotal questions: (a) Are both internal and external sources of knowledge sufficient to tackle the posed question? and (b) Is the reasoning process of the QA LLM susceptible to common issues often encountered in multi-hop QA?

To address these concerns, the evaluating step employs two types of metacognitive knowledge: procedural knowledge and declarative knowledge. Within cognitive psychology [19, 27], procedural knowledge embodies the grasp of methodologies and strategies essential for confronting specific tasks, while declarative knowledge is anchored in specific facts or content-based information, covering facts and concepts associated with problem solving. For Retrieval-augmented LLMs, we convert the role of LLM from the original question answering system to an evaluator-critic system. Different from the question-answering perspective which forces the model to generate an answer, evaluator-critic perspective can more objectively judge the limitations of its reasoning process towards the answer. Below we will introduce how to leverage two types of metacognitive knowledge to answer the above two questions.

**Procedural Knowledge**. This domain of knowledge is crucial for examining the sufficiency of both the internal and external knowledge for a given question. To address question (a), we propose model-based methods to evaluating the answer automatically, simulating human annotators in Sec. 3.2. We leverage the evaluator-critic LLM to gauge the adequacy of its internal knowledge. Meanwhile,

a natural language inference (NLI) model is employed to measure the sufficiency of external knowledge. Note that this process may be affected by the accuracy of the LLM and the NLI model, but can be replaced by any better model in the future.

- *Internal Knowledge Evaluating*: We capitalize on the inherent capacity of the LLM to determine if a question can be aptly answered using its built-in knowledge. To do this, we present the question $q$ to the evaluator-critic LLM, which functions as an evaluator here and offers a binary outcome based on:

$$\text{LLM}_{\text{Eval-Critic}}(q, \text{Prompt}_{\text{Eval}}) \qquad (3)$$

Prompt$_{\text{Eval}}$: Please act as an evaluator-critic system, determine if you can provide a reliable answer to the {question} based on your own knowledge?

- *External Knowledge Evaluating*: To gauge the adequacy of external knowledge sources, we deploy an advanced NLI model TRUE [8] to examine if the retrieved documents, represented as $D_q$, provide enough information to answer the question. The process is formulated as:

$$f\left(\left[\{d_i\}_{i=1}^{|D|}\right], q\right), \qquad (4)$$

with $f(\text{premise}, \text{hypothesis})$ being the function of the NLI model. It returns a value of 1 if the premise entails the hypothesis, otherwise, it returns 0.

Upon evaluating through the aforementioned model-based evaluating methods, we can classify the situation into four categories (as in Sec. 3.2) in an automatic manner: no knowledge; only internal knowledge; only external knowledge; both internal and external knowledge. Each situation highlights specific potential sources of errors, leading to varying strategies employed for future planning depending on the identified category.

**Declarative Knowledge**. Addressing question (b), declarative knowledge within MetaRAG is directed towards identifying prevalent error patterns. This aids in pinpointing possible pitfalls in the reasoning process. We've categorized typical mistakes into three distinct types:

- *Incomplete Reasoning*: This error is the most prevalent in multi-hop QA. It arises when the model fails to utilize all relevant fragments from the given context or does not follow a comprehensive chain of thought to arrive at the correct answer.
- *Answer Redundance*: This pertains to instances where the model delivers an overly verbose or repetitive answer. Such redundancy can arise when the model identifies multiple analogous data points but cannot consolidate them effectively.
- *Ambiguity Understanding*: This error manifests when the model misunderstands the subtleties or nuances embedded within a query, leading it to generate answers based on related but incorrect references.

Each of these mistakes poses distinct challenges when employing LLMs for multi-hop reasoning tasks. Relying on declarative knowledge (DK), we invoke the critic functionality of the evaluator-critic LLM to determine if the proposed answer falls prey to any of these errors. For each error type, we furnish a description and several examples in the format {Error name - Error description - Examples}. Subsequently, the question $q$, documents $D_q$, and answer $y$ are fed into the LLM, functioning as a critic in this context:

$$\text{LLM}_{\text{Eval-Critic}}([\text{DK}, q, D_q, y], \text{Prompt}_{\text{Critic}}) \qquad (5)$$

Prompt$_{\text{Critic}}$:
```
Please act as an evaluator-critic system, assess
whether the {response} based on {references} for
the {question} contains any {error types}?
```

Through the evaluating phase, the model gains an understanding of potential issues within the current answer, which may stem from gaps in knowledge or deficiencies in the reasoning process. Once these challenges are identified, the model can then develop customized solutions to enhance the precision of its reasoning in the context of question-answering. We detail this as follows.

## 4.3 Planning: Strategizing Answer Refinement

In the domain of metacognition, the concept of planning refers to the effective regulation of the original cognitive process, guided by the results obtained from the evaluation stage. Previous empirical studies in Section 3.2 have illuminated three primary challenges in multi-hop QA when the model fails to provide accurate answers: insufficient knowledge, conflicting knowledge, and erroneous reasoning. After identifying the issues during the evaluating stage, in this section, we will introduce planning strategies to address each of these challenges, ensuring a coherent and logical framework to mitigate such hurdles in multi-hop QA scenarios.

**Insufficient Knowledge**. In the first type of condition, there is a lack of both internal and external knowledge to answer the current question. When the evaluator-critic LLM recognizes this situation, it's prompted to **generate a new query** to further retrieve information from the corpus. A well-formulated follow-up query should have two characteristics: (1) It should differ from the original inquiry to specifically target missing information. (2) It should break down the original question into a more specific sub-question. Specifically, given a question $q$, the existing retrieved documents $D_q$, and an answer $y$, we utilize the evaluator-critic LLM's introspective ability to deduce what external knowledge is still lacking:

$$q' = \text{LLM}_{\text{Eval-Critic}}([q, D_q, y], \text{Prompt}_{\text{QG}}), \qquad (6)$$

where Prompt$_{\text{QG}}$ is a kind of instruction that encourages the LLM to ask a new query with "To answer this question, I further need to search {q'}". With this new query $q'$, the model conducts another search to obtain additional documents. These newly retrieved documents are then incorporated into the reference list as new external knowledge.

**Conflicting Knowledge**. Another situation that can result in inaccurate responses is when there's a disparity between internal and external knowledge. When one subset of knowledge is sufficient to answer a question, but another isn't, the model might become confused due to the inconsistency between the two. This scenario can be classified into following two cases:

- *Only Internal Knowledge Available*. When the model is capable of providing the correct answer directly, external references may serve as distractors. For example, if asked about the boiling point of water under standard pressure, an external source may claim it's 93.4 °C at 1,905 metres altitude, leading the model astray. To mitigate this, it's advisable for the model to **discard external references** and rely on its intrinsic knowledge. We achieve this by altering the question-answering prompt, guiding the model to rely solely on its internal knowledge.
- *Only External Knowledge Available*. Conversely, in situations where only external knowledge is present, LLMs can be prone to hallucinations if they mistakenly believe they know the answer. For example, without internal knowledge of a recent event, the model might incorrectly infer details based on similar but outdated events in its training data. To circumvent this, we ask the LLM to **only rely on the provided references** for its response.

**Erroneous Reasoning**. Even if a model can answer questions consistently using both internal and external knowledge, errors may still occur during multi-step reasoning. To address the issue of faulty reasoning, we propose improvements from two perspectives:

- *Double-Checking the Reasoning Process*. First, we aim to verify that each statement in our reasoning process is backed by evidence. To achieve this, we invoke the NLI model $f$ to assess the groundedness of each statement $s_i$ in the LLM output $y$. This will help determine which statements are supported by external references $D_q = \{d_1, d_2, ...\}$ and which ones aren't. Finally, the statements need to be double-checked are:

$$S_{\text{DC}} = \left\{ s_i | f\left( \left[\{d_i\}_{i=1}^{|D|}\right], s_i \right) = 0 \right\}, \qquad (7)$$

For any statement that lacks evidence in $S_{\text{DC}}$, we request the LLM to re-evaluate its correctness, ensuring that the LLM excludes any statement that doesn't meet its confidence threshold.
- *Providing Suggestions*. In response to the common errors identified during the evaluating phase, we ask the evaluator-critic LLM to provide specific suggestions for the question-answering LLM based on the specific error type by " Please generate a statement that offers suggestions to prevent the occurrence of the {error type} in future reasoning

processes". These suggestions serve as guidance during the next round of answer reasoning. It's worth noting that if no common errors are detected, we set a default suggestion "Please think step by step." to guide its reasoning process.

The planning at this stage primarily focuses on systematically reducing the model's error rate when operating under conditions of comprehensive and consistent knowledge.

Through the implementation of metacognitive regulation, the evaluator-critic LLM can monitor, evaluate, and plan the cognitive processes of the question-answering LLM. This level of transparency in cognitive space empowers MetaRAG to comprehend the dimensions of "know whether", "know why", and "know how", thereby enhancing its reasoning correctness.

## 5  EXPERIMENTAL SETUP

### 5.1  Datasets and Evaluation Metrics

To test the ability of our proposed method on multi-hop reasoning, we conduct experiments on two multi-hop question answering datasets: HotpotQA [36] and 2WikiMultiHopQA [7]. These two datasets are all constructed based on Wikipedia documents, allowing us to use the consistent document corpus and retrievers to provide external references for LLMs. Considering the constraints of experimental costs, following [12], we sub-sample 500 questions from the validation set of each dataset for experiments.

For evaluation metrics, at answer-level, we use exact match (EM) to test whether the prediction is consistent with the reference answer. At token-level, following [12], we use token-level F1, precision (Prec.) and recall (Rec.) for comprehensive evaluation.

### 5.2  Baselines

For comparison, we choose two closebook models and four retrieval-augmented models as baselines. **Standard Prompting** [2] directs the LLM to respond to queries without referencing any external content. **Chain-of-Thought** [35] furnishes LLM with examples inclusive of reasoning processes to encourage more thoughtful reasoning. **Standard RAG** [20] employs the query to retrieve multiple documents, and inputs them into LLM for deriving answers. **Re-Act** [37] framework proposes synergizing reasoning and acting in language models, delineating the question-answering process into thought, action, and observation phases. **Self-Ask** [24] integrates intermediate steps to assist in deliberating on complex issues. **Reflexion** [30] incorporates an evaluator to reinforce language agents through linguistic feedback.

To ensure a balanced comparison between MetaRAG and the baseline methodologies, uniform settings are maintained across all models. This encompasses identical in-context demonstrations, prompt formats, retrievers, and document corpora.

### 5.3  Implementation Details

In cognition process, we choose the cutting-edge `gpt-35-turbo-16k` LLM by querying its API[1] iteratively with a temperature setting of 0. Since both datasets predominantly depend on knowledge from Wikipedia, we utilize the Wikipedia dump [14] to serve as the document corpus, where articles are segmented into passages of 100

tokens. The retrieval of relevant documents from this corpus employs the BM25 algorithm [26], selecting the top 5 passages to serve as the external knowledge.

Transitioning to the metacognition process, we leverage a fine-tuned T5-large model[2] which acts as our expert monitoring model. The efficacy of similarity calculations is based on a repository of sentence transformers[3]. We set a default judgment threshold for our monitoring mechanism at 0.4 to ensure precision. The maximum number of iterations is set to 5. The NLI model used in evaluating and planning is entrusted to a T5-XXL model[4].

## 6  RESULTS AND ANALYSIS

### 6.1  Main Results

The main results are shown in Table 1. It can be observed that:

(1) Our proposed MetaRAG consistently surpasses all other baseline methods across two datasets. When compared to the baseline Reflexion, which also integrates a self-critic mechanism in its reasoning process, MetaRAG demonstrates a substantial improvement of over 26.0% in terms of EM on the HotpotQA dataset. This suggests that using a metacognitive strategy is more beneficial than merely relying on self-criticism. By leveraging metacognitive knowledge and regulation, our method aligns better with human thought processes. This allows for a better identification of errors or gaps in knowledge during reasoning, leading to enhanced answer accuracy.

(2) Models equipped with a self-critic mechanism demonstrate superior performance compared to those without it. When compared with the multi-time retrieval baseline Self-Ask, both Reflexion and MetaRAG show improvements of over 6.3% and 34.0% respectively on the HotpotQA dataset. This indicates that by assigning a critic role to LLMs, they gain the ability to assess the quality of their own responses from a different perspective. MetaRAG further considers the conditions of knowledge and the accuracy of multi-hop reasoning, allowing it not only to pinpoint mistakes but also to identify the cause of these mistakes.

(3) Upon comparing two datasets, we observe a more significant improvement with MetaRAG on 2WikiMultihopQA than on HotpotQA, boosting performance by 34.6% and 26.0% respectively when compared to the baseline model Reflexion. Upon closer examination of the datasets, we note that the 2WikiMultihopQA set exhibits a higher proportion of conflicting knowledge, meaning there is a higher incidence where the retriever retrieves information that is inconsistent with the knowledge contained within the LLM. MetaRAG adeptly addresses this by meticulously formulating planning strategies based on varying conditions of knowledge, enhancing the precision of reasoning in a targeted manner.

In summary, our proposed MetaRAG explores the synergy of external and internal knowledge in retrieval-augmented LLMs based on metacognition principles in cognitive science, leading to enhanced accuracy in multi-hop reasoning tasks.

### 6.2  The Study of Monitoring Phase

As a critic step within the metacognitive process, monitoring plays a pivotal role in evaluating the validity of responses generated by

---

[1]https://api.openai.com/v1/chat/completions

[2]https://huggingface.co/gaussalgo/T5-LM-Large_Canard-Fullwiki-HotpotQA
[3]https://huggingface.co/sentence-transformers/all-MiniLM-L6-v2
[4]https://huggingface.co/google/t5_xxl_true_nli_mixture

**Table 1: Evaluation results on two multi-hop question answering datasets. ✓ and − indicates reasoning with and without the retrieval (Retr.), multi-time retrieval (Multi.), and critic component. "†" denotes the result outperforms baseline models in t-test at $p < 0.05$ level. The best results are in bold and the second best results are underlined.**

| Method | Retr. | Multi. | Critic | HotpotQA | | | | 2WikiMultihopQA | | | |
|---|---|---|---|---|---|---|---|---|---|---|---|
| | | | | EM | F1 | Prec. | Rec. | EM | F1 | Prec. | Rec. |
| *Without retrieval (Closebook)* | | | | | | | | | | | |
| Standard Prompting | - | - | - | 20.0 | 25.8 | 26.4 | 28.9 | 21.6 | 25.7 | 24.5 | 31.8 |
| Chain-of-Thought | - | - | - | 22.4 | 34.2 | 33.9 | 46.0 | 27.6 | 37.4 | 35.8 | 44.3 |
| *With retrieval (BM25)* | | | | | | | | | | | |
| Standard RAG | ✓ | - | - | 24.6 | 33.0 | 34.1 | 34.5 | 18.8 | 25.2 | 25.6 | 26.2 |
| ReAct | ✓ | ✓ | - | 24.9 | 41.7 | 42.6 | 44.7 | 21.0 | 28.0 | 27.6 | 30.0 |
| Self-Ask | ✓ | ✓ | - | 28.2 | 43.1 | 43.4 | 44.8 | 28.6 | 37.5 | 36.5 | 42.8 |
| Reflexion | ✓ | ✓ | ✓ | 30.0 | 43.4 | 43.2 | 44.3 | 31.8 | 41.7 | 40.6 | 44.2 |
| MetaRAG (ours) | ✓ | ✓ | ✓ | **37.8**† | **49.9**† | **52.1**† | **50.9**† | **42.8**† | **50.8**† | **50.7**† | **52.2**† |

**Table 2: The comparison of various monitoring expert models with different parameter size (Param.) on 2WikiMultihopQA.**

| Expert model | Param. | EM | F1 | Prec. | Rec. |
|---|---|---|---|---|---|
| *Large Language Models* | | | | | |
| LLaMA2-chat | 13B | 40.4 | 47.6 | 47.6 | 48.8 |
| ChatGLM2 | 6B | 39.8 | 48.8 | 48.5 | 50.5 |
| *Fine-tuned QA Models* | | | | | |
| SpanBert-large | 0.34B | 42.0 | 50.4 | 50.3 | 51.8 |
| T5-large | 0.77B | 42.8 | 50.8 | 50.7 | 52.2 |

LLMs. In order to understand the impact of monitor on the overall framework, we conduct two experiments by comparing different monitoring models and the similarity threshold $k$.

**Monitoring Models Variation.** We begin by evaluating the impact of using various expert models, which determine whether activating metacognition, as monitors. We primarily focus on two categories of models for monitoring: large language models and fine-tuned QA models. LLMs, like LLaMA2-chat [32] and ChatGLM2 [3], offer notable zero-shot capabilities, allowing them to assess answer quality effectively. On the other hand, the fine-tuned QA models, such as SpanBERT-large [13] and T5-large, are smaller but have been specifically trained on particular datasets to become experts in their domains. We provide details on their parameter sizes and compare their performance.

As illustrated in Table 2, utilizing fine-tuned QA models as the expert model during the monitoring phase yields superior performance compared to large language models. This suggests that fine-tuned QA models can offer more precise feedback with fewer parameters, meeting the efficiency and effectiveness requirements of the monitoring stage. The performance of LLaMA2-chat and ChatGLM2 indicates that using LLMs to self-supervise is a feasible approach, setting higher standards for model capabilities. Moreover, the T5-large slightly outperforms SpanBERT-large, which might be attributed to the fact that generative models with larger parameter capacities are more apt for this task than extraction-based models.

**Different similarity thresholds.** Secondly, we focus on the relationship between the similarity threshold $k$ in the monitor and overall performance. This threshold dictates the ease of triggering metacognitive processes. A higher threshold implies a higher likelihood for activating metacognitive evaluating. We test the range of $k$ from 0.2 to 0.8, incrementing by 0.1, and report the metacognitive proportion and answer quality on 2WikiMultihopQA.

As depicted in Figure 4(a), when the threshold is set to 0.2, roughly 15% of questions are directed to the evaluator-critic LLM for metacognitive reasoning. At this point, there is approximately a 20% improvement compared to Reflexion. As the threshold increases, the proportion of questions requiring metacognitive reasoning steadily increases. By the time the threshold reaches 0.8, this arrives to 84%. Interestingly, the performance doesn't increase linearly. The model performs best when the threshold is set at 0.4. This suggests that not all questions benefit from metacognitive reasoning. For some straightforward inquiries, overthinking can be counterproductive. This mirrors human tendencies to some extent: going with one's intuition can be more effective than over-analyzing.

### 6.3 Ablation Studies on Meta-knowledge

The metacognitive evaluating employs metacognitive knowledge (declarative and procedural knowledge) to pinpoint potential mistakes in the reasoning and assessing the completeness of internal and external knowledge. To explore the necessity of these two categories of metacognitive knowledge, we conduct ablation studies by eliminating the assessment of internal or external knowledge for procedural knowledge or exclude a type of common error assessment linked to declarative knowledge.

The findings, depicted in Table 3, highlight that stripping away any facet of metacognitive knowledge detrimentally impacts performance across all evaluation metrics. Among these, the omission of procedural knowledge results in the most pronounced decline in model efficiency. This suggests a heightened importance of understanding the interplay between internal and external knowledge, as this understanding is crucial for the model's strategic planning. Within the procedural knowledge category, it's evident that recognizing external knowledge stands out in terms of importance.

**Table 3: Ablation Studies on Meta-knowledge. *Co.* is the consistency between model evaluating and human annotation.**

| Expert model | EM | F1 | Prec. | Rec. |
|---|---|---|---|---|
| *Procedural Knowledge* (Internal *Co.*=0.76; External *Co.*=0.84) | | | | |
| w/o. Internal | 41.4 | 49.5 | 49.6 | 50.5 |
| w/o. External | 37.4 | 44.9 | 45.1 | 45.9 |
| w/o. All | 30.6 | 36.8 | 37.2 | 37.3 |
| *Declarative Knowledge* | | | | |
| w/o. Incomplete | 41.2 | 49.7 | 49.8 | 51.0 |
| w/o. Redundance | 41.6 | 49.3 | 49.3 | 50.6 |
| w/o. Ambiguity | 41.2 | 50.9 | 51.0 | 51.9 |
| w/o. All | 40.6 | 49.2 | 49.3 | 50.5 |
| MetaRAG | 42.8 | 50.8 | 50.7 | 52.2 |

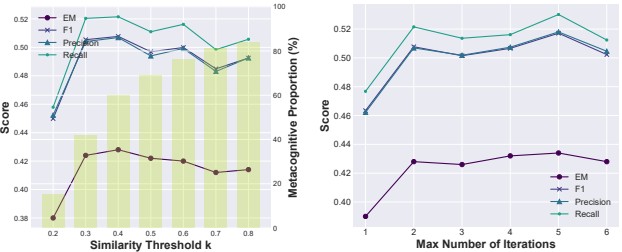

| (a) Similarity Thresholds. | (b) Number of iterations. |
|---|---|

**Figure 4: Performance with different similarity thresholds and the number of iterations on 2WikiMultihopQA.**

This underscores the idea that many questions arise from an insufficiency in external knowledge. Thanks to the architecture of MetaRAG, this deficiency can be alleviated, leading the model to generate new queries for knowledge acquisition. When we turn our focus to declarative knowledge, each type of common error bears significance to the overall model efficacy. Notably, errors stemming from incomplete reasoning seem to be most impactful. This implies that conventional QA prompts cannot effectively harness the multi-hop reasoning abilities of LLMs. However, with the inclusion of metacognitive knowledge, this latent potential can be harnessed, thereby refining the model's reasoning precision.

## 6.4 Performance of Each Planning Strategies

During the planning phase, we employ improvement strategies for three distinct scenarios: insufficient knowledge, conflicting knowledge, and erroneous reasoning. To validate the effectiveness of these strategies, we conduct experiments to measure the enhancements each scenario could offer. We categorize all questions into three scenarios based on the conditions of knowledge, and examine the impact of the planning approach on each scenario.

Figure 5 shows the performance of various models in the three scenarios. Generally, as the richness of knowledge increases, the accuracy of each model improves. The ReAct and Reflexion models enhance the performance in situations of insufficient knowledge through employing multi-time retrieval and critic mechanisms. However, the improvement in scenarios of conflicting knowledge

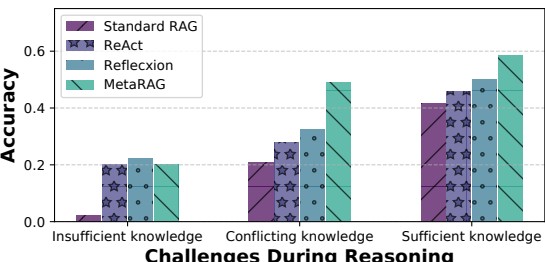

**Figure 5: Performance of the planning strategy under each knowledge conditions: insufficient, conflicting, sufficient.**

and complete knowledge is relatively marginal. In contrast to these two methods, our proposed MetaRAG significantly boosts the accuracy of reasoning in these two scenarios. This achievement is primarily attributed to MetaRAG's meticulous analysis of conflicts between internal and external knowledge and common error types through a metacognitive process, thereby optimizing the model's reasoning process in a targeted manner.

## 6.5 Exploration of the Number of Iterations

In the context of MetaRAG, monitoring is crucial in determining whether to proceed to the next stage of the metacognitive process. It's important to emphasize that the results are significantly influenced by the maximum number of iterations. To identify the ideal number, we systematically increase the maximum iteration count from 1 to 6, while closely observing how the accuracy changes.

As depicted in Figure 4(b), the accuracy of MetaRAG improves progressively as the maximum iteration count increases. However, once the iteration count reaches 5, the performance peaks, indicating that deeper metacognitive reflection can indeed enhance inference accuracy. Nevertheless, excessively increasing the number of reflection rounds leads to a slight decline in results. This could be attributed to the model's diminishing ability to extract more useful information or suggestions through the metacognitive mechanism. An intriguing observation is a minor accuracy peak at an iteration count of 2. This phenomenon primarily arises from the characteristics of the 2WikiMultihopQA dataset, where the majority of questions require references from two sources. Two rounds of metacognitive reflection prove sufficient to gather the necessary knowledge for these questions. Beyond that, additional rounds of reflection tend to introduce noise, resulting in fluctuating results.

## 7 CONCLUSION

In this paper, we proposed MetaRAG, a novel framework combining the retrieval-augmented LLMs process with human-inspired metacognition to enhance multi-hop reasoning. Through a structured metacognitive process involving monitoring, evaluating, and planning stages, MetaRAG facilitates model awareness on its own reasoning process. This empowers the model to identify the sufficiency of knowledge and potential mistakes during reasoning. Experimental results on two multi-hop QA datasets demonstrated the superior performance of MetaRAG over existing baselines. In the future, we aspire to incorporate more human cognitive approaches, including emotional understanding, intuition, and cultural awareness, into the reasoning process of LLM.

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
