# OpenReview forum: "Metacognitive Retrieval-Augmented Large Language Models"
_ACM.org/TheWebConf/2024/Conference — TheWebConf24 Oral_

### Official Review · Reviewer_dEp2 · 2023-11-14

**Novelty:** 4
**Technical Quality:** 5

**Review:**

**Summary**: This paper introduces MetaRAG, which combines retrieval-enhanced generative processes with human-inspired metacognition. The three major steps of monitoring, evaluating, and planning ensure that the model evaluates the adequacy of its answers, identifies the causes of potential shortcomings, and develops a plan for improvement. Empirical evaluation of two datasets shows that MetaRAG significantly outperforms existing methods.

**Pros**:
1. The motivation that comes from the metacognition process is new and interesting. It is just like two agents working together to solve a hard problem.
2. The structure of the paper is clear and easy to understand.
3. The proposed method can be well aligned with the human metacognition processing steps.

**Cons**:
1. The major concern is that in order to fit the process of human metacognition process, the MetaRAG introduces many off-the-shelf models in the framework, e.g. expert model, LLM model, NLI model, and so on. How to make sure the improvement comes from the MetaRAG framework not comes from the additional information in these imported models.
2. Using LLM to act both as a QA system and an evaluator system may lead to some unexpected issues. For example, the QA LLM gives the answer to the question, while the evaluator LLM and NLI module may deny to say it can answer the question. How to deal with this conflict? Is the QA LLM wrong or is the evaluator LLM wrong?
3. Lack of baselines, such as ToolFormer [1], Plan-and-Solve [2], Verify-and-Edit [3], Tree-of-Thought [4], SearChain [5], DSP [6] and so on.
4. The performances of some baselines are significantly lower than the original paper. For example, the Self-Ask method achieves 40.1 EM on the 2Wiki dataset while the authors report 28.6. If the settings are different, please describe these differences in detail in the paper. And I also suggest the authors compare their method in the same settings of Self-Ask whose performance is significantly better than current experimental settings.
5. As shown in Table 3, the performance gain almost comes from External information (Retrieval). The other tricks and the framework contribute less than 1 EM point. It seems this complex human metacognition process does not really work.

[1] Toolformer: Language Models Can Teach Themselves to Use Tools

[2] Plan-and-Solve Prompting: Improving Zero-Shot Chain-of-Thought Reasoning by Large Language Models

[3] Verify-and-Edit: A Knowledge-Enhanced Chain-of-Thought Framework

[4] Tree of Thoughts: Deliberate Problem Solving with Large Language Models

[5] Search-in-the-Chain: Towards the Accurate, Credible and Traceable Content Generation for Complex Knowledge-intensive Tasks

[6] Demonstrate-SearchPredict: Composing retrieval and language models for knowledge-intensive NLP

**Questions:**

The major questions are listed in the Cons.

1. As mentioned in Line 431, the LLM and NLI models are not perfect, how to deal with the errors imported from these models? The same question is the use of expert models.
2. Why only use BM25 as a retriever? Recent off-the-shelf dense retrievers are much stronger than BM25.
3. Line 264 is not clear to me. How to categorize the questions into these four groups? I don't think LLM can know whether it can answer the question just using a prompt. Furthermore, when the LLM can give the answer in the previous step. The logic here is very weird.
4. Line 296-299 "However, existing research on retrieval-augmented LLMs primarily focuses on the design of the reasoning steps, overlooking the awareness of the reasoning itself." This sentence is hard to understand. What is the difference between 'reasoning steps' and 'reasoning itself'?
5. In Section 4.1, the author compares the answers with the expert model to determine whether it is necessary to enter the metacognitive process. If the expert model is recognized as the ground truth of the question, why is MetaRAG needed to answer the question? In other words, is the performance of the MetaRAG ceiling the expert model?
6. Some prompts are provided in Section 4. However, based on my understanding, making simple adjustments to the prompt can significantly impact performance. Have you conducted any searches to ensure that the prompt you are presenting yields better results?
7. In addition to ablation experiments, can you give some cases to prove that the various stages of MetaRAG are effective?

**Reviewer Confidence:**

4: The reviewer is certain that the evaluation is correct and very familiar with the relevant literature

**Scope:**

4: The work is relevant to the Web and to the track, and is of broad interest to the community

---

### Official Review · Reviewer_DHmB · 2023-11-20

**Novelty:** 5
**Technical Quality:** 5

**Review:**

Summary:

The paper introduces the Metacognitive Retrieval-Augmented Generation (MetaRAG) framework, a novel approach that integrates retrieval-augmented generation in large language models (LLMs) with human-inspired metacognition. The authors argue that traditional retrieval-augmented models are limited by predefined reasoning steps, leading to inaccuracies. MetaRAG addresses this issue by enabling the model to self-reflect and critically evaluate its cognitive processes, thereby enhancing introspective reasoning abilities. The framework employs a three-step metacognitive regulation pipeline—monitoring, evaluating, and planning—to assess and refine answers. It has demonstrated significant performance improvements over existing methods in empirical evaluations on multi-hop question-answering datasets.

-----
Strengths:
- MetaRAG’s integration of metacognition into LLMs represents an advanced approach.
- The proposed approach shows significant improvement across several comprehensive datasets.
- The ablation studies are thorough and provide insightful results.
-----
Weaknesses:
- The paper lacks a complexity and cost analysis.
- The adaptability of MetaRAG to tasks beyond multi-hop QA needs further exploration to assess its versatility and generalizability.
- Various other multi-agent and LLM-based systems could have been considered as baselines. While these systems do not explicitly mention the metacognitive retrieval step, their methodologies are similar. Therefore, a better and clearer comparison with related work such as MetaGPT, AutoGPT, LangChain, Autogen, and Multi-Agent Debate would be beneficial. Some of these might have been released after the paper's submission, but they are worth considering for future comparisons.

-----
Comments:
- The authors frequently categorize the reasons behind system failures without experimental support or clarity on the completeness of these reasons. For example, they mention that the main reasons for failure are insufficient knowledge, conflicting knowledge, and erroneous reasoning, but do not detail how they arrived at these conclusions or their validity. Again we have something similar in section 4.2 for typical mistakes.
- I think Figure 2 is quite vague. What is counted on the left axis of the figure? What does the dotted line represent, and what exactly do the bar plots show? Why is ReAct performing worse with additional information? On what set of queries has this figure been experimented?
- The human annotation process described in Section 3.2 is not detailed, and additional information in the appendices would be beneficial.
- Another relevant work for comparison is referenced at https://arxiv.org/abs/2309.11392.
- Figure 3, while visually appealing, is confusing in its content and not self-explanatory.
- It would be insightful if the authors mentioned the cost of running each model in Table 1.
- Extending the analysis beyond six iterations in Figure 4(b) would provide more conclusive insights, as the current one-step approach with no statistical test is insufficient to conclude that more steps confuse the LLMs.

**Questions:**

- How did the authors determine the reasoning behind the models' failures?
- Will the authors release the code? Given the high cost of experiments, releasing experiment logs would be beneficial.
- How does the paper distinguish its work from other recently proposed LLM-based multi-agent systems? ( see comments)
- Can the authors provide more details on Figure 2 and its experimental setup? ( see commments)

**Reviewer Confidence:**

3: The reviewer is confident but not certain that the evaluation is correct

**Scope:**

4: The work is relevant to the Web and to the track, and is of broad interest to the community

---

### Official Review · Reviewer_YLV2 · 2023-11-22

**Novelty:** 5
**Technical Quality:** 4

**Review:**

The paper proposes a metacognition framework to enable self introspection and self improvement of LLM's cognition process for retrieval augmentation. The actual components and techniques are mostly based on zero-shot or few-shots prompting. Experiments are performance on HotpotQA and 2WikiMultiHopQA, by comparing it against other RAG based approach.

Pros
* Motivation is clear and the intuition of metacognition makes sense
* Nice breakdown of different failure patterns: insufficient knowledge, conflicting knowledge and erroneous reasoning.
* The overall framework makes sense and is described clearly
* The results are strong comparing to the listed baselines

Cons
* The approach would increase the inference cost and requirement on context length quite a bit, good to have some discussions on the performance cost tradeoffs
* The expert model part seems strange to me. If there is an expert model, why not just use it for the QA task itself? Basically it assumes a stronger model (the expert model), but not using it. Worth doing an ablating by just removing the expert model (hence the monitoring phase)

**Questions:**

Please address the questions in the Review section. Additionally:
* Would make the paper more interesting to consider the effects while multiple different models are used at different phases, how different models would interact and play out.

Typo:
* line 83, but -> by

**Reviewer Confidence:**

3: The reviewer is confident but not certain that the evaluation is correct

**Scope:**

4: The work is relevant to the Web and to the track, and is of broad interest to the community

---

### Official Review · Reviewer_Jqhp · 2023-11-26

**Novelty:** 4
**Technical Quality:** 5

**Review:**

This paper proposes a Retrieval Augmentation Method and presents experimental results on two QA datasets. While the findings suggest the effectiveness of the proposed approach, I would like to bring some concerns that need to be addressed for the manuscript to reach its full potential.

1. **Limited Experimental Diversity:**
   The experimental evaluation in your study is conducted on only two datasets. Previous works, such as "Active Retrieval Augmented Generation" [1], have performed experiments on a more diverse set of datasets, including a minimum of four. Expanding the experimental scope to include a broader range of datasets would enhance the generalizability and robustness of your proposed method.

2. **Missing State-of-the-Art (SOTA) Baselines:In the RAG (Retrieval-Augmented Generation) field, there are numerous new methods, but the baselines in this paper are not sufficient.**
   This paper lacks a comparison with state-of-the-art baselines, such as FLARE [1], IR-CoT[2], which are relevant to the proposed method. The absence of SOTA baselines may leave readers with an incomplete understanding of the method's competitiveness.

[1] Jiang, Zhengbao, et al. "Active retrieval augmented generation." arXiv preprint arXiv:2305.06983 (2023).
[2] Harsh Trivedi, Niranjan Balasubramanian, Tushar Khot, and Ashish Sabharwal.2022. Interleaving retrieval with chain-of-thought reasoning for knowledge-intensive multi-step questions. arXiv preprint arXiv:2212.10509 (2022

**Questions:**

Please refer to the previous section

**Ethics Review Description:**

No such issue

**Reviewer Confidence:**

3: The reviewer is confident but not certain that the evaluation is correct

**Scope:**

3: The work is somewhat relevant to the Web and to the track, and is of narrow interest to a sub-community

---

### Decision · Program_Chairs · 2024-01-22

**Decision:**

Accept (Oral)

**Comment:**

This paper proposes a retrieval augmentation method and presents experimental results on two QA datasets.

 The paper was reviewed by four reviewers. The paper has clearly some merits. The reviewers agree on the technical quality and novely of the papers, but they also raise some comments still requirint a propert explanation. Please clarify this point in the camera-ready copy.